# Power Positioning System Control Study of "Intelligent Research and Internship Vessel" Based on Terminal Sliding Mode

Zhenghao Wei, Zhibin He *, Xiaoyu Wu and Qi Zhang

Marine Engineering College, Dalian Maritime University, Dalian 116026, China;
w1220043662@dlmu.edu.cn (Z.W.); wuxiaoyu@dlmu.edu.cn (X.W.); 1120211172tnxn@dlmu.edu.cn (Q.Z.)
* Correspondence: hezb999@126.com

**Abstract:** As most of the current dynamic positioning systems are based on model ships, they cannot accurately reflect the motion state, position changes, and mutual influence of each part of the dynamic positioning system of actual ships in complex environments. Other actual ships such as cargo ships cannot add various sensors and auxiliary equipment to verify and analyze the positioning system. This article takes the intelligent research and training dual-use ship of Dalian Maritime University, which integrates scientific research and training, as the object of study. This ship will not be affected by the voyage period and route and can choose a suitable sea area for research. Therefore, in order to improve the accuracy and reliability of the dynamic positioning system, research on the ship's dynamic positioning system was carried out. Firstly, an accurate mathematical model was developed to simulate ship motion, focusing on the use of the Dalian Maritime University's intelligent and practical training dual-purpose vessel as the modeling object. Through this approach, a more detailed understanding of the effects of actual environmental perturbations on ship control and positioning can be obtained, as well as more realistic ship control and positioning results. The hydrodynamic derivatives of ship model motion were obtained by numerical calculation and applied to the three-degree-of-freedom model of the intelligent research and training dual-use ship. Then, the model was used as part of the closed-loop simulation model of the ship's dynamic positioning system, and the terminal sliding mode controller was used for simulation and emulation, thereby obtaining ideal simulation test results. Our results deepen the understanding of DPS accuracy and are consistent with the theory of terminal slip modes for ship power positioning control systems. This has implications for improving the accuracy of ship power positioning systems, as previously discussed in previous authors. In conclusion, this study not only improves the accuracy and reliability of the DPS but also proposes the use of the terminal slip film for a ship power positioning control system modeled on the Dalian Maritime University intelligent and practical dual-purpose vessel. These contributions are significant in improving the efficiency, safety, and environmental sustainability of ship operations.

**Keywords:** ship power positioning; hydrodynamic derivatives; intelligent research and practical training dual-use vessels; mathematical modeling of ship motion; simulation; terminal sliding mode

## 1. Introduction

While the development of intelligent ships is progressing, challenges such as improving the accuracy of ship power positioning remain unresolved. This study aims to address these by improving the accuracy of the ship's power positioning control system using the Dalian Maritime University's dual-use ship for intelligence and practical training. The intelligent research and training dual-purpose ship carries the scientific research and innovative practical teaching tasks of the intelligent ships and provides a research platform for the disciplines of traffic information engineering and control, ship and ocean engineering,

electrical engineering, information and communication engineering, control science, and engineering. The research platform is mainly used to overcome the problems of marine scientific research and shipping science and technology, serve the construction of "double first-class", actively build a research base for high-tech ships in the era of artificial intelligence, provide strong scientific and technological support for the construction of a strong maritime nation, and cultivate high-quality shipping talents. When the ship is operating at sea, the wind, waves, and currents on the sea surface have a great influence on the interference load generated by the ship; in order to better achieve the intelligent research and training dual-purpose ship's "intelligent research + student teaching and training" function, the reliability of the ship and control accuracy has higher requirements. In this paper, on the basis of the previous research on ship power positioning, we take the "intelligent research and training dual-use ship" as the research object, calculate the hydrodynamic derivatives, calculate the M inertia coefficient matrix and D damping coefficient matrix, derive the ship's three-degree-of-freedom model, and bring the model to the linear sliding mode and the terminal sliding mode controller for simulation.

Dynamic positioning system (DPS) is a technology that uses ship propellers and control systems to keep ships in a fixed position or on a predetermined course at sea. DPS has a wide range of applications in ocean engineering, offshore operations, and maritime rescue. Although DPS has been explored in existing studies such as Jaros, K, 2022 [1], there still exist gaps such as the study being on modeled or non-intelligent ships, and there are many limitations to the study, which this study aims to fill. The current DPS has some problems, such as low control accuracy, high energy consumption, and sensitivity to environmental interference. In order to solve these problems and improve the performance and reliability of DPS, more in-depth theoretical research and experimental verification are needed. Taking the intelligent research and training dual-use ship of Dalian Maritime University as the object of study, research on ship DPS control was carried out. On the one hand, it can promote the innovation and development of intelligent ship technology, explore and verify more efficient, stable, and energy-saving DPS control algorithms, improve the control accuracy and adaptability of DPS, reduce the energy consumption and environmental impact of DPS, and provide theoretical support and experimental basis for the innovation and development of intelligent ship technology. On the other hand, it can enhance the ability and level of ship intelligent research and experimentation, build a ship intelligent research and experimentation verification platform, conduct research on intelligent navigation technology and system, ship remote monitoring, shore-based support, ship intelligent communication technology, ship intelligent operation and maintenance technology, and other aspects, carry out collaborative research with other intelligent ships, explore the collective behavior and collaborative control of intelligent ships, and enhance China's international influence and competitiveness in the field of intelligent ships. In addition, it can also cultivate high-level talents in the field of intelligent ships, be used to train intelligent shipping talents, serve students' cognitive learning, practical training, and intelligent training, enable students to come into contact with the most advanced intelligent ship equipment and systems and master the basic principles and operation methods of intelligent ships, cultivate students' innovative thinking and practical ability, and cultivate professional talents for the application and development of DPS. Secondly, research on actual ships has many advantages over research on model ships. Actual ship tests can more realistically simulate the motion and control of ships in the marine environment without being limited by factors such as the size, material, and proportion of model ships. These factors affect the hydrodynamic similarity between model ships and actual ships, resulting in model ship test results that need to be corrected and converted for application to actual ships. Actual ship tests can also verify and optimize the results of model ship tests and improve the reliability and performance of ship dynamic positioning systems. The ship dynamic positioning system is a system that controls power to maintain the position and heading of the ship, and it plays an important role in ship control. Actual ship tests can also examine the adaptability and safety of ships in complex natural environments,

as well as the structural strength and durability of ships. Ship performance can be fully tested at full scale under real working conditions, avoiding the differences and uncertainties between model ship test results and actual ship data, and improving the accuracy and credibility of predictions. Model ship testing is a commonly used method for evaluating ship performance, but it has some limitations, such as scale effects, Reynolds number effects, and viscosity effects, which result in some deviation between model ship test results and actual ship data. Therefore, research on actual ships can more realistically reflect the motion state and position changes of ships, external environmental interference, and the interaction and influence of various components of the dynamic positioning system, thereby improving the accuracy and reliability of the dynamic positioning system. Compared with other actual ships, the cargo ship of Dalian Maritime University is used for commercial purposes and cannot add various sensors and auxiliary equipment to verify and analyze the positioning system. This ship will not be affected by the voyage period and route and can choose a suitable sea area for research.

## 2. Intelligent Research and Practical Training Dual-Use Vessel of Dalian Maritime University

Dalian Maritime University's intelligent research and training ship is a dual-purpose ship with the function of "intelligent ship research + students' teaching and training". It is a dual-purpose ship that is focused on national strategic needs, leading the integration of cutting-edge technology of intelligent ships, systematic testing, and experimental research of related equipment, as well as the training of talents in the new industry of intelligent shipping. Each time, it can arrange 30 students for practical training and five researchers for scientific research tests, plus the number of crew members [2] to ensure the normal operation of the ship, which belongs to the special-purpose ship category. This ship not only meets the needs of teaching and practical training but also has the more important function of becoming an open research platform to study ship intelligence and unmanned ships. Through the operation practice of teaching and students' practical training carried out in the real environment at sea, the degree of intelligence, reliability, and stability of intelligent systems, equipment, and technologies such as intelligent system architecture, intelligent technology, and equipment, intelligent integration platform, etc., are examined and verified. The ship has good wind resistance, rapidity, and endurance, flexible maneuverability, and is capable of carrying out normal teaching, practical training, and scientific research work under Class IV sea state and safe navigation under Class VI sea state. The ship is an independently designed and constructed dual-use ship for intelligent research and practical training, with an unlimited design navigation area, an overall length of 69.83 meters, a beam of 10.9 meters, a depth of 5.0 meters, a design draught of 3.5 meters, a design speed of 18 knots, an endurance of about 2500 nautical miles, and a self-sustaining power of 6 days. The ship can accommodate 15 crew members and 30 students. Considering the needs of scientific research, three cabins are set up for scientific researchers to use, and there are five long-term scientific researchers on board [3–14].

## 3. Mathematical Modelling of Ship Power Positioning Systems

*3.1. Ship Power Positioning Systems*

The ship's dynamic positioning system (DP system) is a technology that enables the ship to maintain a specific position and direction at sea by utilizing the propulsion system and steering gear on the ship, as well as the corresponding sensors and computer control systems. The main purpose of the ship's dynamic positioning system is to ensure that the ship can hover stably at designated geographical coordinates in harsh sea conditions, deep-sea operations, or situations where precise position control is required.

In terms of control technology, dynamic positioning systems began to rise in the 1960s. In the 1960s, the first generation of dynamic positioning ships appeared. These usually used conventional control methods and also adopted some auxiliary signal processing

methods, using filters to eliminate high-frequency components in the deviation signal to avoid responding to high-frequency movements.

In the 1970s, researchers such as Balchen combined optimal control with the Kalman filter theory. A new control technology was rapidly applied in ship dynamic positioning systems and began to become the mainstream method of dynamic positioning control. Then, the second modern dynamic positioning system developed rapidly and became the most widely used dynamic positioning system. It is a control technology based on modern control theory and is also the symbol of the second-generation ship dynamic positioning system. The second-generation dynamic positioning control method is currently the most mature system used internationally [2,15–19].

In recent years, research on newer and more advanced control technologies has further developed. Robust control, fuzzy control, neural network control technology, etc., have become increasingly mature. The third-generation dynamic positioning system has begun to adopt intelligent control theories and methods based on intelligent control methods and real-time measurements. Using technology to improve the accuracy of positioning systems is a trend in various countries researching dynamic positioning control, and countries are stepping up research and development. Intelligent control methods are mainly reflected in robust control, fuzzy control, nonlinear model predictive control, etc., [20].

### 3.2. Establishment of a Coordinate System

Firstly, the coordinates are established in the northeast coordinate system and the hull coordinate system, respectively. The vector equation of the ship's position and bow direction is $\eta = [x, y, \varphi]^T$, and the velocity vector equation of the ship is $v = [u, v, r]^T$, as shown in Figure 1 [21].

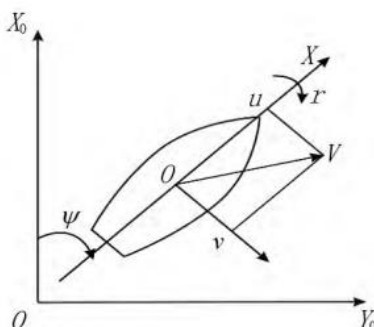

**Figure 1.** Ship planar motion variables.

### 3.3. Build a Mathematical Model of the Ship's Motion

The ship power positioning system, which is mainly studied in this paper, focuses on the positioning control of the ship on the sea surface, so the effects of vertical oscillation, transverse motions, and longitudinal rocking are neglected, and only the three degrees of freedom of longitudinal oscillation, transverse oscillation, and bow rocking are considered. The ship's motion in the horizontal plane is superimposed by the ship's low-frequency and high-frequency motion. The low-frequency motion affects the real position change of the ship, which is the main research object. The simplified three-degree-of-freedom ship low-frequency motion model is obtained [22]:

$$\begin{bmatrix} \dot{x} \\ \dot{y} \\ \dot{\psi} \end{bmatrix} = \begin{bmatrix} \cos\psi & -\sin\psi & 0 \\ \sin\psi & \cos\psi & 0 \\ 0 & 0 & 1 \end{bmatrix} \begin{bmatrix} u \\ v \\ r \end{bmatrix} \Rightarrow \dot{\eta} = R(\psi)v, \tag{1}$$

$$M\dot{v} + D(v)v = \tau + d \tag{2}$$

where $M$ is the mass matrix and $D$ is the damping matrix, $M = \begin{bmatrix} m - X_{\dot{u}} & 0 & 0 \\ 0 & m - Y_{\dot{v}} & 0 \\ 0 & 0 & I_z - N_{\dot{r}} \end{bmatrix}$,

$D(v) = \begin{bmatrix} -X_u & 0 & 0 \\ 0 & -Y_v & 0 \\ 0 & 0 & -N_r \end{bmatrix}$. $X_u, X_{\dot{u}}, Y_v, Y_{\dot{v}}, Y_r, Y_{\dot{r}}, N_v, N_{\dot{v}}, N_r, N_{\dot{r}}$ is the hydrodynamic derivative, $m$ is the mass of the ship, $I_z$ is the rotational moment of inertia of the ship, $\tau$ is the total thrust of the propellers, and $d$ is the ambient force of the ocean. Due to the extremely complex and irregular geometry of the attached ship including oars and rudders, it is unlikely that the hydrodynamic theoretical method can be used to carry out the computation of this series of hydrodynamic conductors. Most of the cases are used in the experimental ship modeling, and the formulae used for the estimation of the hydrodynamic conductors of the linear fluid are adopted here. The experimental data of the ship modeling, which was sorted out by Clarke, are shown below [1,23–26]:

$$I_z = \left(1 - C_b^{4.5}\right)\left(\frac{D}{24g}\right)\left(L^2 + B^2\right) \tag{3}$$

$$X'_u = -SC_t\left(\frac{1}{Ld_m 0.5\rho V^2 L^2}\right) \tag{4}$$

$$X'_{\dot{u}} = -\frac{1}{100}\left[0.398 + 11.97C_b\left(1 + 3.73\frac{d_m}{B}\right) - 2.89C_b\frac{L}{B}\left(1 + 1.13\frac{d_m}{B}\right) + 0.175C_b\left(\frac{L}{B}\right)^2\left(1 + 0.541\frac{d_m}{B}\right) - 1.107\frac{L}{B}\frac{d_m}{B}\right]\left(\frac{m}{0.5\rho L^3}\right) \tag{5}$$

$$Y_{\dot{v}} = -\left[1 + \frac{0.16C_b B}{d_m} - 5.1\left(\frac{B}{L}\right)^2\right]\cdot\pi\left(\frac{d_m}{L}\right)^2 \tag{6}$$

$$Y'_r = -\left[\frac{0.67B}{L} - 0.0033\left(\frac{B}{d_m}\right)^2\right]\cdot\pi\left(\frac{d_m}{L}\right)^2 \tag{7}$$

$$N'_i = -\left(\frac{1.1B}{L} - \frac{0.41B}{d_m}\right)\cdot\pi\left(\frac{d_m}{L}\right)^2 \tag{8}$$

$$N'_{\dot{r}} = -\left(\frac{1}{12} + \frac{0.017C_b B}{d_m} - \frac{0.33B}{L}\right)\cdot\pi\left(\frac{d_m}{L}\right)^2 \tag{9}$$

$$Y'_v = -\left(1 + \frac{0.40C_b B}{d_m}\right)\cdot\pi\left(\frac{d_m}{L}\right)^2 \tag{10}$$

$$Y'_r = -\left(-\frac{1}{2} + \frac{2.2B}{L} - \frac{0.080B}{d_m}\right)\cdot\pi\left(\frac{d_m}{L}\right)^2 \tag{11}$$

$$N'_v = -\left(\frac{1}{2} + \frac{2.4d_m}{L}\right)\cdot\pi\left(\frac{d_m}{L}\right)^2 \tag{12}$$

$$N'_r = -\left(\frac{1}{4} + \frac{0.039B}{d_m} - \frac{0.56B}{L}\right)\cdot\pi\left(\frac{d_m}{L}\right)^2, \tag{13}$$

where $L$ is the hull waterline length, $d_m$ is the average draft depth; $B$ is the ship's breadth; and $C$ is the hull squareness coefficient [27].

### 3.4. The Main Parameters of the Ship

The research object chosen in this paper is the intelligent research and training dual-use ship of Dalian Maritime University. The simulation model is built in Simulink, and its main parameters are shown in Table 1.

**Table 1.** Main parameters of intelligent ship.

| Parameter | Numeric Value |
|---|---|
| Ship length $L$/m | 69.213 m |
| Ship width $B$/m | 10.90 m |
| Draft $d_m$/m | 3.5 m |
| Weight load $m$/kg | 1500 t |
| Displacement $\nabla$/t | 1450 t |
| Design speed $Vd$/kn | 18.0 kn |

The mass matrix and damping matrix of the intelligent research and practical training dual-use vessel of Dalian Maritime University, the object of the study, are, respectively,

$$M = \begin{pmatrix} 0.94 & 0 & 0 \\ 0 & 1.81 & -0.9 \\ 0 & -0.9 & 16.4 \end{pmatrix} \tag{14}$$

$$D = \begin{pmatrix} 0.0181 & 0 & 0 \\ 0 & -0.0136 & -0.064 \\ 0 & -0.0051 & -0.00231 \end{pmatrix} \tag{15}$$

*3.5. Mathematical Models of Environmental Disturbances*

When a ship is traveling and operating at sea, it is often affected by external environmental factors, mainly from the disturbance of wind, waves, and currents. These disturbances will change the ship's position and bow direction, so the calculation and study of wind, wave, and current disturbances are indispensable in the study of the ship's dynamic positioning. In this paper, mathematical modeling of wind, waves, and currents will be carried out.

3.5.1. Wind Disturbance Model

Let the absolute wind speed be $V_T$, the absolute wind angle be $\psi_T$, the relative wind speed be $V_R$, the relative wind angle be $\psi_R$, $V$ is the ship's speed, $u_R$ and $v_R$ are the components of the relative wind speed in the kinematic coordinate system, $\psi$ is the bow angle, and $\beta$ is the drift angle, and the relationship between the relative wind and the wind pressure and moments can be expressed as [28]

$$V_R = V_T - V \tag{16}$$

$$u_R = u + V_T \cos(\psi_T - \psi) \tag{17}$$

$$v_R = -v + V_T \sin(\psi_T - \psi) \tag{18}$$

$$V = \sqrt{u^2 + v^2}, \quad V_R = \sqrt{u_R{}^2 + v_R{}^2} \tag{19}$$

It can also be expressed as

$$\begin{cases} V_R \cos\psi_R = V_T \cos(\psi_T - \psi) - V\cos\beta \\ V_R \sin\psi_R = V_T \sin(\psi_T - \psi) + V\sin\beta \end{cases} \tag{20}$$

$$\beta = tan^{-1}\left(-\frac{v}{u}\right) \tag{21}$$

After the calculation, we obtain

$$V_R^2 = V_T^2 + V^2 + 2V_T V \cos(\psi_T - \beta) \tag{22}$$

Wind disturbance dynamics are expressed as $\omega_{wind} = [X_{wind} Y_{wind} N_{wind}]$; for the sum of

average wind speed and gusts:

$$
\begin{cases}
X_{wind} = \overline{X}_{wind} + \tilde{X}_{wind} \\
Y_{wind} = \overline{Y}_{wind} + \tilde{Y}_{wind} \\
N_{wind} = \overline{N}_{wind} + \tilde{N}_{wind}
\end{cases}, \tag{23}
$$

where the average wind and moment are expressed as

$$
\begin{cases}
\overline{X}_{wind} = \frac{1}{2}C_X(\psi_T)\rho_a V_R^2 A_T & \text{(N)} \\
\overline{Y}_{wind} = \frac{1}{2}C_Y(\psi_T)\rho_a V_R^2 A_L & \text{(N)} \\
\overline{N}_{wind} = \frac{1}{2}C_N(\psi_T)\rho_a V_R^2 A_L & \text{(N m)}
\end{cases} \tag{24}
$$

In the formula, $C_X$, $C_Y$, $C_N$ are wind force (moment) coefficients, $\rho_a$ is air density (kg/m$^3$), $A_T$ is the side projected area of the ship subjected to the wind (m$^2$), $A_L$ is the side projected area of the ship subjected to the wind (m$^2$), $L$ is the total length of the ship (m), and $V_R$ is the relative wind speed (kn).

3.5.2. Wave Perturbation Model

Most of the analyses of wave motion are analyzed using the wave energy spectrum formula, which is used in this paper [28]:

$$
S(w) = Aw^{-5}exp\left(-Bw^{-4}\right) \tag{25}
$$

Calculation of wave disturbance force and torque:

$$
\begin{cases}
X_w = \frac{1}{2}\rho g L \zeta_D^2 C_{XD}(\lambda)\cos\gamma \\
X_w = \frac{1}{2}\rho g L \zeta_D^2 C_{YD}(\lambda)\sin\gamma \\
N_w = \frac{1}{2}\rho g L \zeta_D^2 C_{ND}(\lambda)\sin\gamma
\end{cases}, \tag{26}
$$

where $\zeta_p$ is the average wave amplitude, $\lambda$ is the wavelength, and $C_{XD}(\lambda)$, $C_{YD}(\lambda)$ and $C_{ND}(\lambda)$ are the coefficients of wave force (moment) along the $x$ and $y$ directions and around the $z$-axis. The wave force (moment) coefficient is calculated as follows:

$$
\begin{cases}
C_{XD} = 0.05 - 0.2\left(\frac{\lambda}{L}\right) + 0.75\left(\frac{\lambda}{L}\right)^2 - 0.51\left(\frac{\lambda}{L}\right)^3 \\
C_{YD} = 0.46 - 6.83\left(\frac{\lambda}{L}\right) - 15.65\left(\frac{\lambda}{L}\right)^2 + 8.84\left(\frac{\lambda}{L}\right)^3 \\
C_{ND} = -0.11 + 0.68\left(\frac{\lambda}{L}\right) - 0.79\left(\frac{\lambda}{L}\right)^2 + 0.21\left(\frac{\lambda}{L}\right)^3
\end{cases} \tag{27}
$$

3.5.3. Flow Perturbation Model

The components of the flow velocity in the $x$- and $y$-axes in the kinematic coordinate system are $u_c$ and $v_c$, and can be expressed as

$$
\begin{cases}
u_c = V_c\cos(\psi_c - \psi) \\
v_c = V_c\sin(\psi_c - \psi)
\end{cases} \tag{28}
$$

$V_c$ is the magnitude of the flow velocity in a fixed coordinate system and $\psi_c$ is the direction of ocean currents.

The forces and moments of the currents are expressed as [28]

$$
\begin{cases}
X_c = \frac{1}{2}\rho A_{fw} V_c^2 C_X \\
Y_c = \frac{1}{2}\rho A_{sw} V_c^2 C_Y \\
N_c = \frac{1}{2}\rho A_{fw} L V_c^2 C_N
\end{cases}, \tag{29}
$$

where $C_X, C_Y$ are the coefficients of current force; $C_N$ is the coefficient of current moment; $L$ is the length of the ship; $A_{sw}, A_{fw}$ are the lateral and frontal projected areas of the ship under the waterline; $\beta$ is the drift angle; $X_c, Y_c$ are the longitudinal and transversal oscillatory forces; $N_c$ is the bow-rocking moment; $V_c$ is the current velocity; and $\rho$ is the density of seawater.

## 4. Experiments and Discussions

### 4.1. Sliding Form Controller for Ship Dynamic Positioning System

A dynamic positioning system is a closed-loop control system; using various types of sensors to measure the ship's motion status and position changes, as well as the size and direction of the external wind, waves, currents, and other disturbing forces, the control system controls the ship's thrust device to generate the appropriate thrust and torque in order to offset the disturbing forces so that the ship can maintain the target position and the port direction. The control system is the core of the DPS; the traditional control method of the ship power positioning system does not meet the present ship positioning accuracy. This paper adopts the power positioning linear and terminal sliding mode controller designed based on the sliding film control theory, as well as the combination of Kalman filtering and a terminal sliding mode controller, to carry out the simulation by bringing in the motion model of the dual-purpose intelligent research and practical training ship of Dalian Maritime University.

#### 4.1.1. Sliding Mode Control

Sliding mode control (SMC) is a control theory based on modern control theory, the main mathematical core of which is the Lyapunov function. The core idea of sliding mode control is to establish a sliding mode surface and pull the controlled system to the sliding mode surface so that the system moves along the sliding mode surface. One of the advantages of sliding mode control is that it disregards external perturbations and uncertain parameters and uses a more "violent" approach to control. The essence of sliding mode control is to design the sliding mode surface through systematic errors so that the system can reach a steady state quickly and the robustness of the system can be greatly improved through the adjustment of the switching function [29].

#### 4.1.2. Terminal Sliding Mode Control

Terminal sliding mode control (TSMC) is a control theory method commonly used for stability and robustness control of nonlinear systems. The goal of terminal sliding mode control is to achieve stable control of the system by guiding the system state trajectory to a specific surface called the terminal sliding mode.

It is well known that certain non-smooth characteristics in dynamics can be utilized to achieve superior performance. For instance, research by M. Zak in 1988 demonstrated that introducing a terminal attractor (i.e., first-order dynamics with fractional power) in neural learning rules results in finite-time convergence. Inspired by this, the concept of terminal sliding mode control (TSMC) was proposed in 1992 by S. T. Venkataraman and S. Gulati for second-order systems and later extended to handle higher-order single-input single-output (sIsO) systems and a class of multi-input multi-output (MIMO) systems in the studies of X. Yu and Z. Man, among others. Leveraging the finite-time convergence property, TSMC, as highlighted by M. Zak, can amplify control forces to expedite convergence [13].

In the early 21st century, a series of theoretical breakthroughs were achieved. Y. Feng, X. Yu, and Z. Man [14] introduced the nonsingular TSMC to address singularity issues in TSMC systems. Additionally, a fast TSMC was developed to accelerate the convergence of basic TSMC for higher-order SISO systems. Continuous TSMC was also introduced for robot control. Since then, the theory and applications of TSMC have gained significant momentum, experiencing substantial growth in publications, with an annual increase of over 20% in the past five years according to Google Scholar's data [14].

4.1.3. Terminal Sliding Mode Control Design for Dynamic Positioning of Dual-Purpose Ship for Intelligent Research and Training

The mathematical model of the ship with three degrees of freedom derived from the previous is

$$\begin{cases} \dot{\eta} = R(\psi)v \\ M\dot{v} + Dv = \tau + d \end{cases} \tag{30}$$

Let the error of the system be $e_2$ and the speed error be $\dot{e}_2$, to obtain

$$\begin{cases} e_2 = \eta - \eta_d \\ \dot{e}_2 = \dot{\eta} - \dot{\eta}_d \end{cases} \tag{31}$$

Combined with the three-degree-of-freedom mathematical model, the following can be obtained:

$$\ddot{\eta} = \left(MR^{-1}\right)^{-1}\left(\tau - \dot{\eta}\left(M\dot{R}^{-1} + DR^{-1}\right)\right) \tag{32}$$

Define the end die face:

$$s_2 = \dot{e}_2 + \lambda sig(e_2)^a \tag{33}$$

Deriving $s_2$ gives:

$$\begin{aligned} \ddot{s}_2 &= \ddot{e}_2 + \lambda\left[\alpha|e_{21}|^{a-1}\dot{e}_{21}, \cdots, \alpha|e_{2n}|^{a-1}\dot{e}_{2n}\right]^T \\ &= \left(MR^{-1}\right)^{-1}\left(\tau - \dot{\eta}\left(M\dot{R}^{-1} + DR^{-1}\right)\right) - \ddot{\eta}_d + \\ &\quad \lambda\left[\alpha|e_{21}|^{a-1}\dot{e}_{21}, \cdots, \alpha|e_{2n}|^{a-1}\dot{e}_{2n}\right]^t \end{aligned} \tag{34}$$

When $s_2 = 0$, the equivalence is

$$\begin{aligned} \tau_{20} &= \ddot{\eta}_d\left(MR^{-1}\right) + \dot{\eta}\left(M\dot{R}^{-1} + DR^{-1}\right) - \\ &\quad \lambda\left(MR^{-1}\right)\left[a|e_{21}|^{a-1}\dot{e}_{21}, \cdots, a|e_{2n}|^{a-1}\dot{e}_{2n}\right]^T \end{aligned} \tag{35}$$

The switching control law is

$$r\tau_{21} = -\left(MR^{-1}\right)\left(k_2\frac{s_2}{\| s_2 \|_2 + \delta} + \xi_2 s_2\right) \tag{36}$$

Therefore, the synovial control law of the dynamic positioning terminal of the ship is obtained:

$$\tau_2 = \tau_{20} + \tau_{21} \tag{37}$$

Based on this control law, the controller model is designed as follows in Figure 2. The * in the figure is the multiplication sign. Modeling using matlab, version number R2020b [30,31].

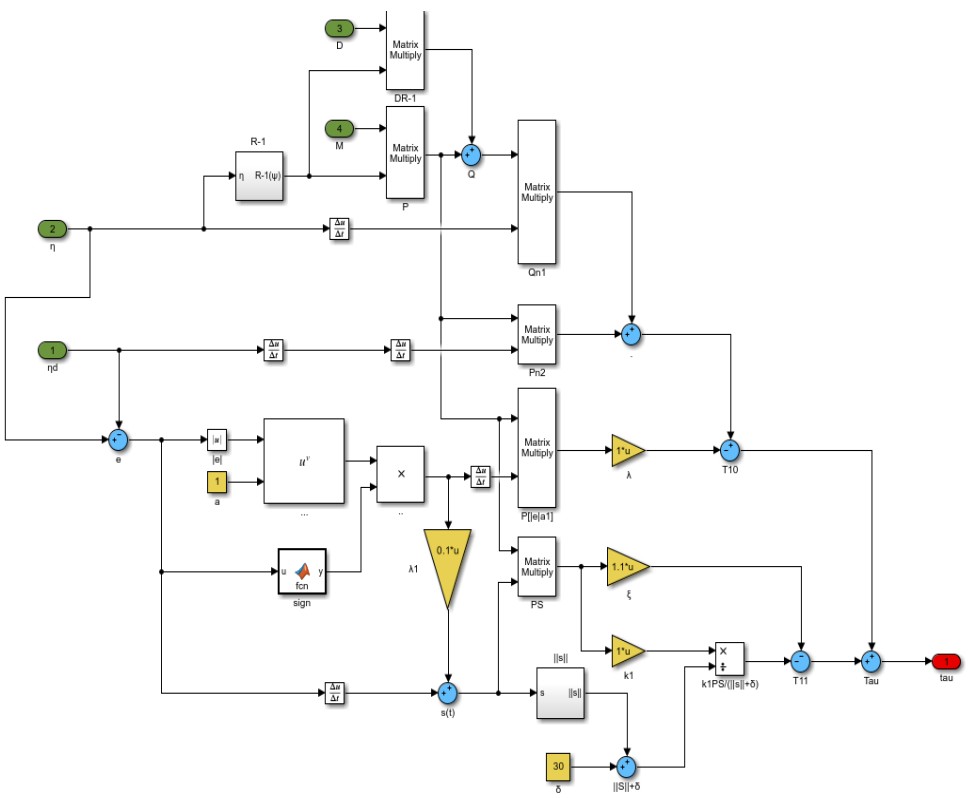

**Figure 2.** Terminal sliding mode controller simulation model.

4.1.4. Simulation of Terminal Slip Film Controller for Power Positioning of Intelligent Research and Training Dual Purpose Vessels

The three-degree-of-freedom model of the dual-use ship for intelligent research and training at Dalian Maritime University is combined with the terminal sliding film controller to build the ship power positioning terminal sliding mode control system in Figure 3. The * in the figure is the multiplication sign [15]

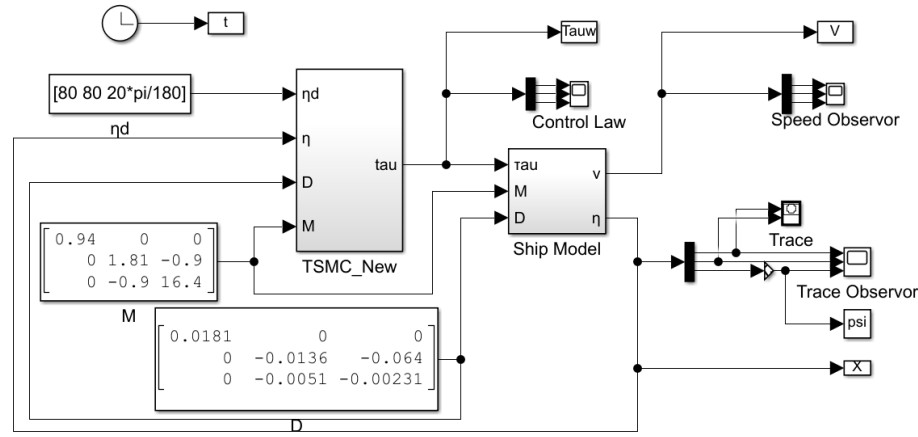

**Figure 3.** Dynamic positioning terminal sliding mode control system simulation model.

The dual-purpose ship for intelligent research and training of Dalian Maritime University was used as the simulation object to carry out the simulation. The initial position of the ship was set to [0 m, 0 m, 0°], and the desired position was set to [80 m, 20 m, 10°]. The parameter settings of the controller were $\mathcal{C}_1 = 1, k_1 = 5, \xi_1 = 3$, and $\delta = 30$. The simulation time was 600 s.

From Figure 4, it can be seen that the speed of the ship gradually increases from 0 and then gradually decreases, indicating that the speed control of controlled positioning

is ideal. From Figure 5, it can be seen that the ship's position reaches the desired position from the initial position after a period of time under the control of the terminal sliding film controller, and the process of controlled positioning is precise, smooth, and fast. From Figure 6, it can be seen that the control force emitted by the ship's control system is ideal, continuous, and bounded. The simulation results of the three figures show the ideality and feasibility of the terminal slip film controller.

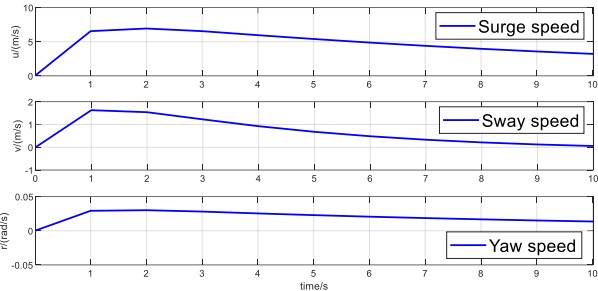

**Figure 4.** Curves of ship speed.

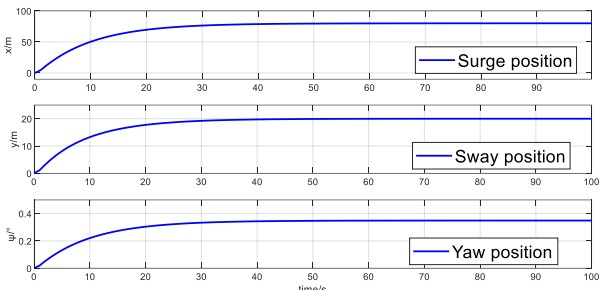

**Figure 5.** Ship position and heading angle.

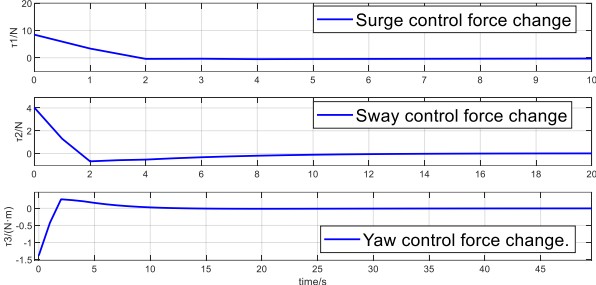

**Figure 6.** Curves of ship control forces.

In order to further explore the performance of the sliding film controller of the power positioning terminal of a dual-use vessel for intelligent research and practical training at Dalian Maritime University, the parameters of the controller were set to $\mathcal{C}_1 = 1$, $k_1 = 5$, $\xi_1 = 3$, $\delta = 30$, $\lambda = 0.1$, $k_2 = 1$, $\xi_2 = 1.1$, $\delta = 30$, $a = 0.5$, and $k = 10$. The initial position of the ship is set to [0 m, 0 m, 0°], the desired position is set to [80 m, 20 m, 10°], and the simulation time is set to 100 s. A comparison of the simulation using the saturation function and the symbolic function is made to obtain the change in the control force of longitudinal oscillation, transverse oscillation position, bow rocking angle, and the change in the control force of the three. The saturated function has a better simulation effect and greatly reduces the vibration of longitudinal oscillation, transverse oscillation, and bow rocking control force, as can be seen in Figures 7–12. The 饱和函数 in the graph is saturation function, The 符号函数 in the graph is symbolic function.

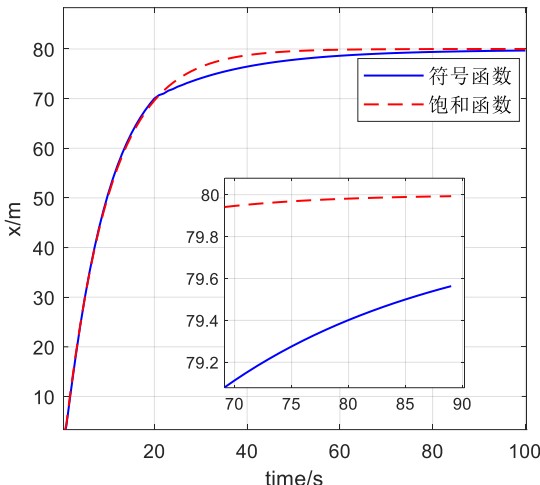

**Figure 7.** Surge position change.

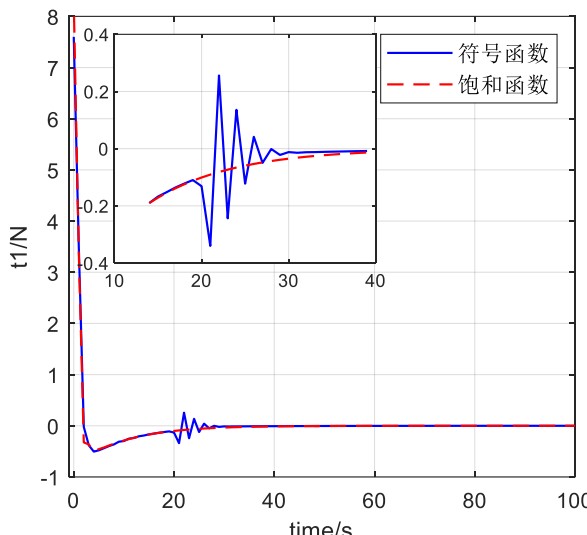

**Figure 8.** Surge control force change.

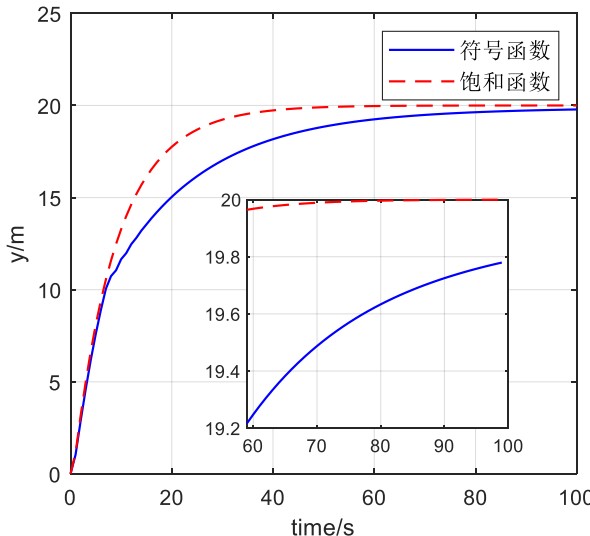

**Figure 9.** Sway position change.

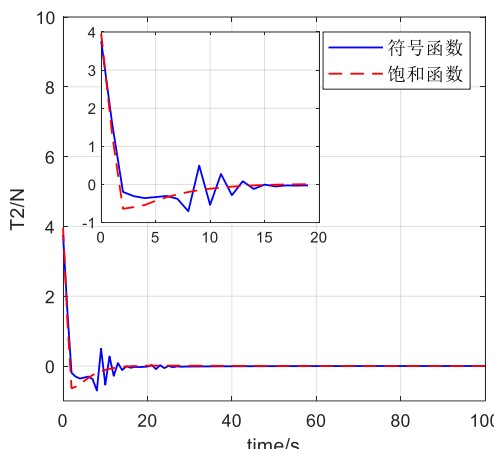

**Figure 10.** Sway control force change.

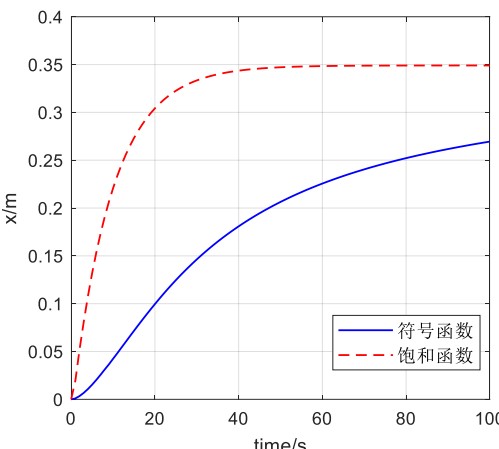

**Figure 11.** Yaw position change.

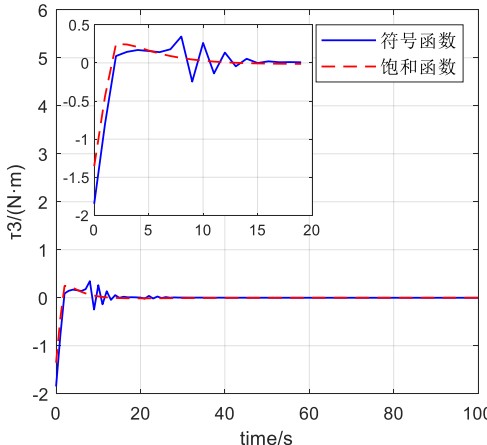

**Figure 12.** Yaw control force change.

During the study, it was found that the parameter $\delta$ has a great influence on the control effect of the controller and other parameters have insignificant eeffects so $\delta$ = 0.3, 3, and 30 were set to demonstrate the performance of the controller. The simulation time was 600 s.

Figures 13–15 show the variation in the ship control force, and analyzing the images, it can be understood that the larger $\delta$ is, the smaller the jitter of the controller is. The more precise control force, as shown in Figures 16–18, demonstrates the variation in the ship's position in the three degrees of freedom of longitudinal oscillation, transverse oscillation, and bow rocking. From the figure, it can be understood that when $\delta$ is larger, the ship

can reach the specified position more quickly and more precisely, and the jitter is reduced. Figures 19–21 show the variation in the ship speed of longitudinal oscillation, transverse oscillation, and bow rocking. There were changes in the ship's longitudinal swing, transverse swing, and bow rocking speed, and from the figure, it can be seen that when δ is smaller, the ship's speed change situation is drastic, and the vibration of the speed change is drastic.

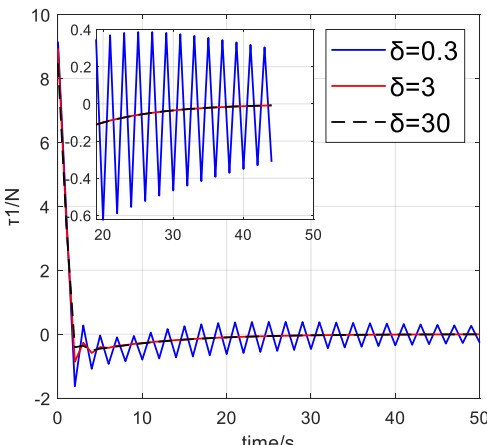

**Figure 13.** Surge control force change.

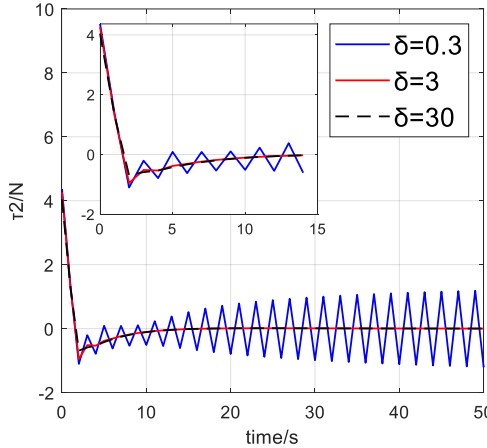

**Figure 14.** Sway control force change.

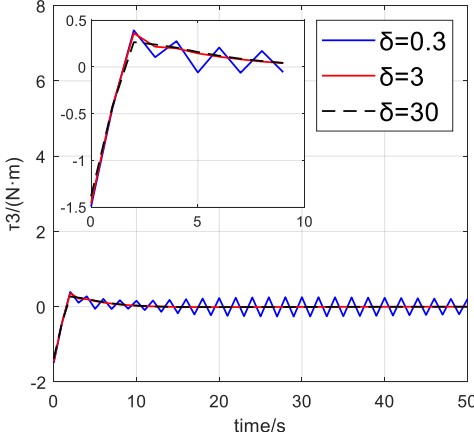

**Figure 15.** Yaw control torque change.

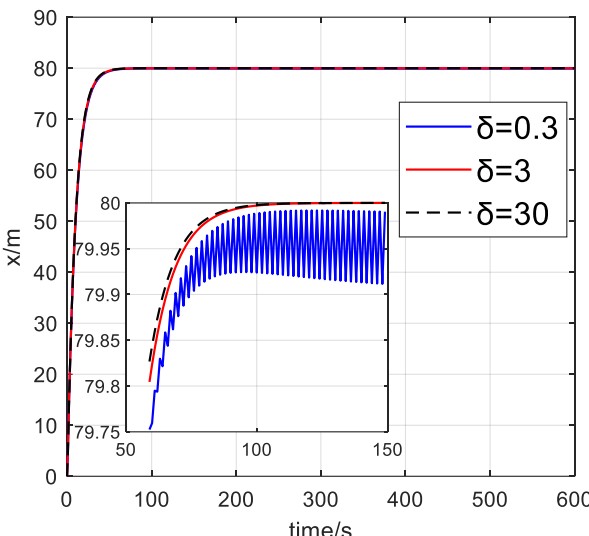

**Figure 16.** Surge position change.

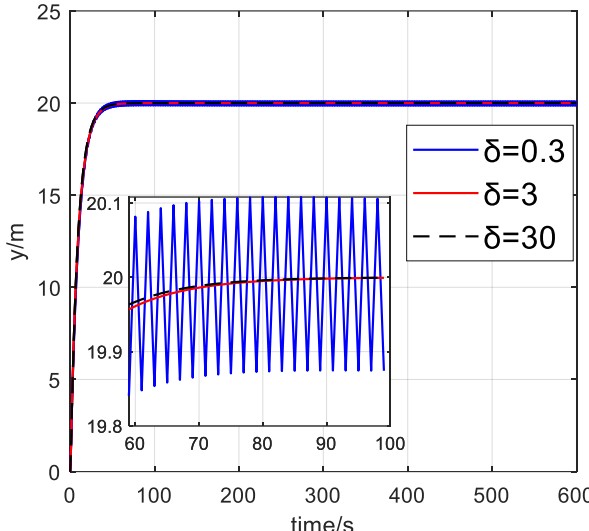

**Figure 17.** Sway position change.

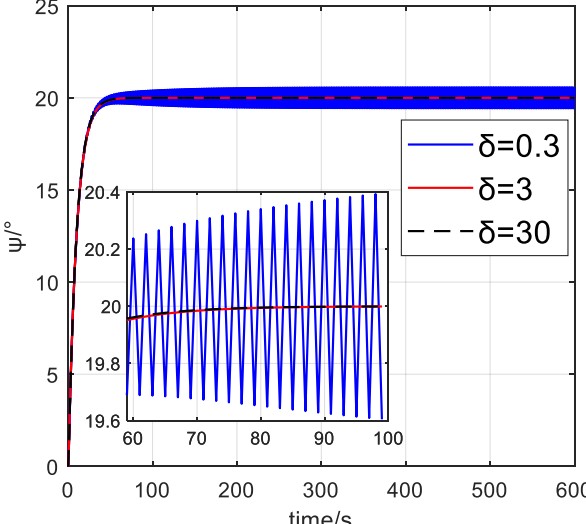

**Figure 18.** Yaw angle change.

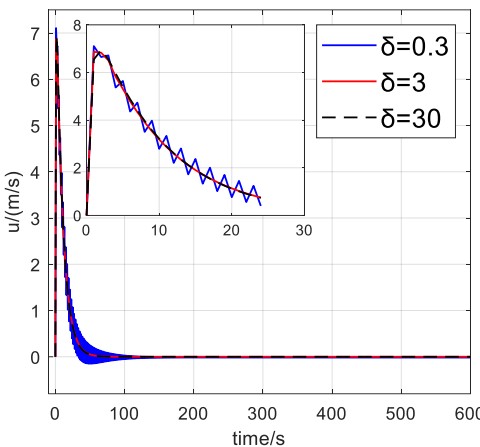

**Figure 19.** Surge speed change.

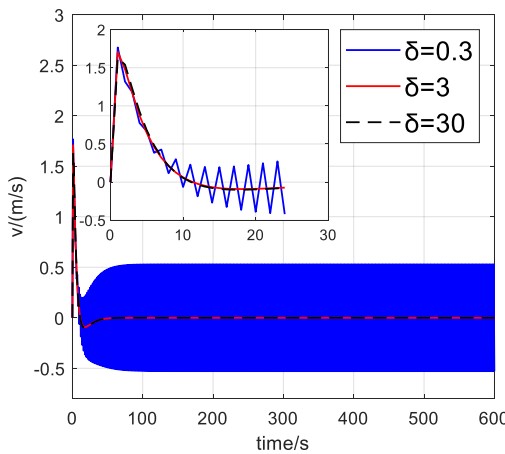

**Figure 20.** Sway speed change.

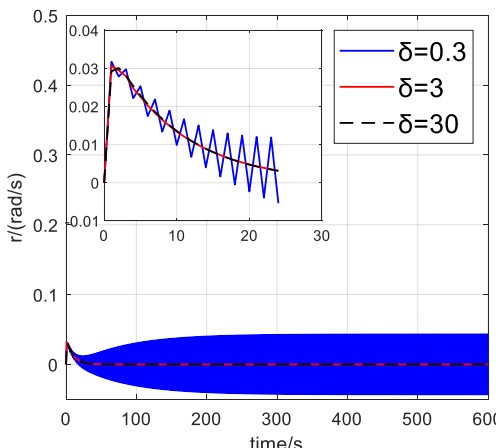

**Figure 21.** Yaw speed change.

## 5. Conclusions

This study conducted an in-depth discussion and research on the dynamic positioning system of an "Intelligent Research and Internship Vessel" based on the terminal sliding mode theory. Through the application of the terminal sliding mode control method, we have achieved the following important research results: Through terminal sliding mode control, the dynamic positioning system of the "Intelligent Research and Internship Vessel" shows remarkable stability under different sea states, working conditions, and robustness. This provides a reliable control guarantee for ships to perform tasks in complex ocean en-

vironments. Drawing on the characteristics of finite time convergence, we successfully created a system reaching the target position and attitude within a limited time. This is of great significance for improving the operational efficiency of the ship and the speed of task execution. Through the introduction of terminal sliding mode control, we have effectively optimized the performance of the "Intelligent Research and Internship Vessel" dynamic positioning system. This not only improves the response speed of the system but also enhances the system's resistance to external disturbances and uncertainties. And the impact of adjusting controller parameters on controlling the ship was further studied through simulation comparison. In future research, we recommend further exploration and optimization of the application of terminal sliding mode control methods in dynamic positioning systems to adapt to a wider range of maritime tasks and environments. In addition, the integration of emerging technologies and sensors is also an important direction to improve system intelligence and performance.

Through the efforts of this research, we believe that the dynamic positioning system of the "Intelligent Research and Internship Vessel" will play an important role in future ocean engineering and research missions. This research provides valuable contributions to the fields of marine science and technology, providing new control methods and theoretical support for safer, more efficient, and sustainable offshore operations.

**Author Contributions:** Conceptualization, Z.W.; funding acquisition, Z.H.; investigation, X.W.; methodology, Z.W.; project administration, Z.W.; software, Z.W.; validation, Q.Z.; writing—original draft, Z.W.; writing—review and editing, Z.W. All authors have read and agreed to the published version of the manuscript.

**Funding:** This research was funded by Ministry of Industry and Information Technology Project: Innovation Project of the Offshore LNG Equipment Industry Chain: CBG3N21-2-7.

**Institutional Review Board Statement:** Not applicable.

**Informed Consent Statement:** Not applicable.

**Data Availability Statement:** The data presented in this study are available on request from the corresponding author. The data are not publicly available due to privacy.

**Acknowledgments:** The authors extend their appreciation to the anonymous reviewers for their valuable feedback.

**Conflicts of Interest:** The authors declare no conflicts of interest.

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
