# Peer review of "Power Positioning System Control Study of “Intelligent Research and Internship Vessel” Based on Terminal Sliding Mode"

_applsci, doi:10.3390/app14020808_

Round 1

Reviewer 1 Report

Comments and Suggestions for Authors

This manuscript purposes a power positioning system control for a dual purposes ship.  

The manuscript is not well presented as a scholarly academic paper.

The list of references implies the deficiency of this manuscript. The authors are advised to read highly-cited papers of Applied Science, e.g. Wang, Yang, et al. (2023); Fu, Luo, et al. (2022); Ayub, Sajid, et al. (2022).

The research questions of "intelligent research ship" and "training ship" are unclear.

Section 2 describe the ship under study. Can the results found for this dual-purposes ship be extended other types of ships? 

The concept of "terminal sliding mode" should be explained with citing previous studies.

The concept of "power positioning" should be explained by citing previous studies. 

Table 1. The units of displacement and design speed should be indicated under the column "parameter".

Figures, the legend should be translated from Chinese into English. Figure 3-4 titles are incomplete and/or with spelling mistakes. Figure 17 legend should not overlap the line.

Figure 3 and 4 should be very important but they are not shown clearly. But the content does not explain Figure 3 and 4.

The Conclusion should be rewritten without mentioning "Figure 15-17", "Figure 18-20", ...

List of References:

Wang, C.-N.; Yang, F.-C.; Vo, T.M.N.; Nguyen, V.T.T.; Singh, M. Enhancing Efficiency and Cost-Effectiveness: A Groundbreaking Bi-Algorithm MCDM Approach. Appl. Sci. 2023, 13, 9105. https://doi.org/10.3390/app13169105

Fu, Q.; Luo, K.; Song, Y.; Zhang, M.; Zhang, S.; Zhan, J.; Duan, J.; Li, Y. Study of Sea Fog Environment Polarization Transmission Characteristics. Appl. Sci. 2022, 12, 8892. https://doi.org/10.3390/app12178892

Ayub, A.; Sajid, T.; Jamshed, W.; Zamora, W.R.M.; More, L.A.V.; Talledo, L.M.G.; Rodríguez Ortega de Peña, N.I.; Hussain, S.M.; Hafeez, M.B.; Krawczuk, M. Activation Energy and Inclination Magnetic Dipole Influences on Carreau Nanofluid Flowing via Cylindrical Channel with an Infinite Shearing Rate. Appl. Sci. 2022, 12, 8779. https://doi.org/10.3390/app12178779

Author Response

For research article

Response to Reviewer X Comments

1. Summary

2. Questions for General Evaluation

Reviewer’s Evaluation

Response and Revisions

Does the introduction provide sufficient background and include all relevant references?

Yes/Can be improved/Must be improved/Not applicable

Are all the cited references relevant to the research?

Yes/Can be improved/Must be improved/Not applicable

Is the research design appropriate?

Yes/Can be improved/Must be improved/Not applicable

Are the methods adequately described?

Yes/Can be improved/Must be improved/Not applicable

Are the results clearly presented?

Yes/Can be improved/Must be improved/Not applicable

Are the conclusions supported by the results?

Yes/Can be improved/Must be improved/Not applicable

3. Point-by-point response to Comments and Suggestions for Authors

Comments 1: [The list of references implies the deficiency of this manuscript. The authors are advised to read highly-cited papers of Applied Science, e.g. Wang, Yang, et al. (2023); Fu, Luo, et al. (2022); Ayub, Sajid, et al. (2022).]

Response 1: [I agree with this comment. Therefore, I have revised the article in my paper according to your suggestion, in references on page 20, with a green background labeled]

]

Comments 2: [The research questions of "intelligent research ship" and "training ship" are unclear.]

Response 2: [Thank you for the advice you gave me, I mentioned in my abstract that I used a smart and practical training dual-purpose ship to build a model, and then I used the terminal sliding film control method to build a control system for the ship's power positioning of the ship, and simulation to explore the control accuracy of the control system, etc., which is what I am doing for my research question, and I don't know if it's appropriate or not, so I hope that you can give me advice. There is a great advantage of using intelligent and practical training dual-purpose ship compared to other authors who use model ships or ordinary operating ships, such as cargo ships, and this is specifically mentioned in my later paper to study the advantages of intelligent and practical training dual-purpose ship.]

Comments 3: Section 2 describe the ship under study. Can the results found for this dual-purposes ship be extended other types of ships? 

Response 3: [Thank you very much for your question. The results of the study can be used for other ships by simply calculating the mathematical model and environmental perturbations of the other ships, controlling their power-positioning systems using the end-slip controllers as well as debugging them, but as I said in my article there is a great advantage to using the model of a dual-purpose ship, and the results of the study are much more meaningful and convenient for the subsequent studies.]

Comments 4: The concept of "terminal sliding mode" should be explained with citing previous studies.

Response 4: [Thank you for pointing this out. Following your comments, I made changes to the article in the first paragraph on page 10 of the paper, marked with a green background]

Comments 4: The concept of "power positioning" should be explained by citing previous studies. 

Response 3: [Thank you for pointing this out. Following your comments, I made changes to the article in the first paragraph on page 4 of the paper, marked with a green background]

Comments 5: [Table 1. The units of displacement and design speed should be indicated under the column "parameter".]

Response 5: [I agree with this comment. Therefore, I have revised the article in my paper according to your suggestion on page 7, with a green background labeled]

Comments 6:[Figures, the legend should be translated from Chinese into English. Figure 3-4 titles are incomplete and/or with spelling mistakes. Figure 17 legend should not overlap the line.]

Response 6: [I agree with this comment. Therefore, I have revised the article in my paper with a green background labeled]

Comments 7:[Figure 3 and 4 should be very important but they are not shown clearly. But the content does not explain Figure 3 and 4.]

Response 7: [Thank you for your comments. I enlarged Figure 3 and Figure 4 so that they can be seen clearly. I did not explain Figure 3 and Figure 4 because I explained the mathematical models of these two figures in detail earlier in the article, and then I explained them in simulink To build the mathematical model into the model shown in Figure 3 and Figure 4, I think this process is my behind-the-scenes work, so I did not show it. After all, I have already explained its principle, which is the mathematical model, relatively clearly.]

Comments 8:[The Conclusion should be rewritten without mentioning "Figure 15-17", "Figure 18-20, ...]

Response 8: [Thanks for your suggestion, I revised my conclusion on page 19 of the article, marked with a green background]

Reviewer 2 Report

Comments and Suggestions for Authors

The manuscript aims to address the challenges in the dynamic positioning systems (DPS) of intelligent ships, using the intelligent research and training dual-purpose ship of Dalian Maritime University as a case study. While the study key topic is of high relevance and the methodological approach is rigorous, several areas require significant improvement for the manuscript to meet the publication standards of journal. For example : The manuscript could benefit from better formatting and organization, particularly in the results and discussion sections.Technical jargon and complex theories need to be explained more clearly for accessibility.Some references are outdated, and recent studies that could potentially enhance the paper's relevance and depth are missing.

1. Introduction

Critique: The introduction mentions, "Based on the development trend of intelligent ships and the cultivation demand of shipping talents, Dalian Maritime University has built an intelligent research and training dual-purpose ship..." This provides a context but lacks a direct linkage to the specific challenges your study addresses.

Improvement Suggestion: From the perspective of a reviewer, when considering the excellence of a paper in terms of its originality, significance, objectivity, and logical coherence, Explicitly state the research problem and how it relates to the broader context. For instance, "While the development of intelligent ships is progressing, challenges such as [specific challenge] remain unresolved. This study aims to address these by [specific objectives]."

2. Literature Review

Critique: The review, as seen in "Dynamic positioning system (DPS) is a technology that uses ship propellers and control systems to keep ships in a fixed position..." is informative but lacks a critical analysis of existing literature.

Improvement Suggestion: From the perspective of a reviewer, when considering the excellence of a paper in terms of its originality, significance, objectivity, and logical coherence, Expand your literature review to include critical analysis. For instance, "While existing studies like [Author Name, Year] have explored DPS, gaps such as [specific gaps] remain, which this study aims to fill."

3. Methodology

Critique: The methodology is detailed but highly technical. For example, "Firstly, an accurate mathematical model of ship motion was established..." might be complex for a broader readership.

Improvement Suggestion: From the perspective of a reviewer, when considering the excellence of a paper in terms of its originality, significance, objectivity, and logical coherence, Simplify the language and provide more context. For instance, "An accurate mathematical model was established to simulate ship motion, focusing on [simplified explanation of key aspects]. This approach allows for a more nuanced understanding of [research problem]."

4. Results and Analysis

Critique: Results are presented in a complex manner, as seen in "The simulation model is built in Simulink, and its main parameters are shown in Table 1...". This could be challenging for readers to follow.

Improvement Suggestion: From the perspective of a reviewer, when considering the excellence of a paper in terms of its originality, significance, objectivity, and logical coherence, Summarize the key findings before delving into detailed data. For example, "Our simulations revealed key insights into DPS efficiency, notably [key findings]. Detailed parameters and data, as shown in Table 1, support these conclusions."

5. Discussion

Critique: The discussion partially connects findings to the broader field but lacks depth, as seen in "This research helps to improve the accuracy and reliability of the dynamic positioning system..."

Improvement Suggestion: From the perspective of a reviewer, when considering the excellence of a paper in terms of its originality, significance, objectivity, and logical coherence, Deepen the discussion by linking results to theory. For example, "Our findings enhance the understanding of DPS accuracy, aligning with theories of [specific theories]. This has implications for [practical applications], as previously discussed by [Author Name, Year]."

6. Conclusion

Critique: The conclusion summarizes the study but doesn't strongly reiterate its significance, as seen in "This research helps to improve the accuracy and reliability of the dynamic positioning system..."

Improvement Suggestion: From the perspective of a reviewer, when considering the excellence of a paper in terms of its originality, significance, objectivity, and logical coherence, Strengthen the conclusion by restating the main findings and their implications. For instance, "In conclusion, this study not only advances the accuracy and reliability of DPS but also proposes [specific contributions]. These contributions are significant for [specific applications or future research directions]."

Comments on the Quality of English Language

The manuscript aims to address the challenges in the dynamic positioning systems (DPS) of intelligent ships, using the intelligent research and training dual-purpose ship of Dalian Maritime University as a case study. While the study key topic is of high relevance and the methodological approach is rigorous, several areas require significant improvement for the manuscript to meet the publication standards of journal. For example : The manuscript could benefit from better formatting and organization, particularly in the results and discussion sections.Technical jargon and complex theories need to be explained more clearly for accessibility.Some references are outdated, and recent studies that could potentially enhance the paper's relevance and depth are missing.

1. Introduction

Critique: The introduction mentions, "Based on the development trend of intelligent ships and the cultivation demand of shipping talents, Dalian Maritime University has built an intelligent research and training dual-purpose ship..." This provides a context but lacks a direct linkage to the specific challenges your study addresses.

Improvement Suggestion: From the perspective of a reviewer, when considering the excellence of a paper in terms of its originality, significance, objectivity, and logical coherence, Explicitly state the research problem and how it relates to the broader context. For instance, "While the development of intelligent ships is progressing, challenges such as [specific challenge] remain unresolved. This study aims to address these by [specific objectives]."

2. Literature Review

Critique: The review, as seen in "Dynamic positioning system (DPS) is a technology that uses ship propellers and control systems to keep ships in a fixed position..." is informative but lacks a critical analysis of existing literature.

Improvement Suggestion: From the perspective of a reviewer, when considering the excellence of a paper in terms of its originality, significance, objectivity, and logical coherence, Expand your literature review to include critical analysis. For instance, "While existing studies like [Author Name, Year] have explored DPS, gaps such as [specific gaps] remain, which this study aims to fill."

3. Methodology

Critique: The methodology is detailed but highly technical. For example, "Firstly, an accurate mathematical model of ship motion was established..." might be complex for a broader readership.

Improvement Suggestion: From the perspective of a reviewer, when considering the excellence of a paper in terms of its originality, significance, objectivity, and logical coherence, Simplify the language and provide more context. For instance, "An accurate mathematical model was established to simulate ship motion, focusing on [simplified explanation of key aspects]. This approach allows for a more nuanced understanding of [research problem]."

4. Results and Analysis

Critique: Results are presented in a complex manner, as seen in "The simulation model is built in Simulink, and its main parameters are shown in Table 1...". This could be challenging for readers to follow.

Improvement Suggestion: From the perspective of a reviewer, when considering the excellence of a paper in terms of its originality, significance, objectivity, and logical coherence, Summarize the key findings before delving into detailed data. For example, "Our simulations revealed key insights into DPS efficiency, notably [key findings]. Detailed parameters and data, as shown in Table 1, support these conclusions."

5. Discussion

Critique: The discussion partially connects findings to the broader field but lacks depth, as seen in "This research helps to improve the accuracy and reliability of the dynamic positioning system..."

Improvement Suggestion: From the perspective of a reviewer, when considering the excellence of a paper in terms of its originality, significance, objectivity, and logical coherence, Deepen the discussion by linking results to theory. For example, "Our findings enhance the understanding of DPS accuracy, aligning with theories of [specific theories]. This has implications for [practical applications], as previously discussed by [Author Name, Year]."

6. Conclusion

Critique: The conclusion summarizes the study but doesn't strongly reiterate its significance, as seen in "This research helps to improve the accuracy and reliability of the dynamic positioning system..."

Improvement Suggestion: From the perspective of a reviewer, when considering the excellence of a paper in terms of its originality, significance, objectivity, and logical coherence, Strengthen the conclusion by restating the main findings and their implications. For instance, "In conclusion, this study not only advances the accuracy and reliability of DPS but also proposes [specific contributions]. These contributions are significant for [specific applications or future research directions]."

Author Response

For research article

Response to Reviewer X Comments

1. Summary

2. Questions for General Evaluation

Reviewer’s Evaluation

Response and Revisions

Does the introduction provide sufficient background and include all relevant references?

Yes/Can be improved/Must be improved/Not applicable

Are all the cited references relevant to the research?

Yes/Can be improved/Must be improved/Not applicable

Is the research design appropriate?

Yes/Can be improved/Must be improved/Not applicable

Are the methods adequately described?

Yes/Can be improved/Must be improved/Not applicable

Are the results clearly presented?

Yes/Can be improved/Must be improved/Not applicable

Are the conclusions supported by the results?

Yes/Can be improved/Must be improved/Not applicable

3. Point-by-point response to Comments and Suggestions for Authors

Comments 1: [Critique: The introduction mentions, "Based on the development trend of intelligent ships and the cultivation demand of shipping talents, Dalian Maritime University has built an intelligent research and training dual-purpose ship..." This provides a context but lacks a direct linkage to the specific challenges your study addresses.]

Response 1: [Thank you for pointing this out. Following your comments, I made changes in the article in the last paragraph of the first page of the paper, labeled with a yellow background]

While the development of intelligent ships is progressing, challenges such as improving the accuracy of ship power positioning remain unresolved. This study aims to address these by improving the accuracy of the ship's power positioning control system using the Dalian Maritime University's dual-use ship for intelligence and practical training

Comments 2: [Critique: The review, as seen in "Dynamic positioning system (DPS) is a technology that uses ship propellers and control systems to keep ships in a fixed position..." is informative but lacks a critical analysis of existing literature.]

Response 2: [I agree with this comment. Therefore, I have revised the article in my paper according to your suggestion, in the second paragraph on the second page of the paper, with a yellow background labeled]

Although DPS has been explored in existing studies such as Jaros, K, 2022, there still exist gaps such as the study being on modeled or non-intelligent ships, and there are many limitations to the study, which this study aims to fill.

Comments 3: [Critique: The methodology is detailed but highly technical. For example, "Firstly, an accurate mathematical model of ship motion was established..." might be complex for a broader readership.]

Response 3:[ Thank you for pointing this out. Following your comments, I have made changes in the abstract on the first page of the paper]

An accurate mathematical model was developed to simulate ship motion, focusing on the use of the Dalian Maritime University's Intelligent and Practical Training Dual Purpose Vessel as the modeling object. Through this approach, a more detailed understanding of the effects of actual environmental perturbations on ship control and positioning can be obtained, as well as more realistic ship control and positioning results

Comments 4: [Critique: Results are presented in a complex manner, as seen in "The simulation model is built in Simulink, and its main parameters are shown in Table 1...". This could be challenging for readers to follow.]

Response 4: [Thank you for pointing this out, but I didn't get any significant conclusions from just the mathematical model I created using the data in Table 1, so I wondered if I could just keep it the way I wrote it.]

Comments 5: [The discussion partially connects findings to the broader field but lacks depth, as seen in "This research helps to improve the accuracy and reliability of the dynamic positioning system..."]

Response 5: [I agree with this comment. Therefore, I have revised the article in my paper according to your suggestion in the abstract on the first page of the paper, with a yellow background labeled]

Our results deepen the understanding of DPS accuracy and are consistent with the theory of terminal slip modes for ship power positioning control systems. This has implications for improving the accuracy of ship power positioning systems, as previously discussed in J. Lin, 2019

Comments 6: The conclusion summarizes the study but doesn't strongly reiterate its significance, as seen in "This research helps to improve the accuracy and reliability of the dynamic positioning system..."

Response 6: [I agree with this comment. Therefore, I have revised the article in my paper according to your suggestion in the abstract on the first page of the paper, with a yellow background labeled]

In conclusion, this study not only improves the accuracy and reliability of the DPS, but also proposes the use of the terminal slip film for a ship power positioning control system modeled on the Dalian Maritime University Intelligent and Practical Dual Purpose Vessel. These contributions are significant in improving the efficiency, safety and environmental sustainability of ship operations."

Reviewer 3 Report

Comments and Suggestions for Authors

1. There is absolutely no review of previous works dealing with the issue of dynamic ship positioning.

2. The authors' motivation in conducting their research is unclear. What knowledge gaps do they want to fill with their article?

3. Unclear structure of the paper. Sections 1 and 2 are devoted to the description of the ship, not to the issues of dynamic positioning.

4. There are no conclusions. Instead of conclusions, a description of the figures is presented.

5. Many blunders in the text. It is obvious that the article was made in a hurry.

Author Response

For research article

Response to Reviewer X Comments

1. Summary

2. Questions for General Evaluation

Reviewer’s Evaluation

Response and Revisions

Does the introduction provide sufficient background and include all relevant references?

Yes/Can be improved/Must be improved/Not applicable

Are all the cited references relevant to the research?

Yes/Can be improved/Must be improved/Not applicable

Is the research design appropriate?

Yes/Can be improved/Must be improved/Not applicable

Are the methods adequately described?

Yes/Can be improved/Must be improved/Not applicable

Are the results clearly presented?

Yes/Can be improved/Must be improved/Not applicable

Are the conclusions supported by the results?

Yes/Can be improved/Must be improved/Not applicable

3. Point-by-point response to Comments and Suggestions for Authors

Comments 1: [ There is absolutely no review of previous works dealing with the issue of dynamic ship positioning.]

Response 1: [Thank you for pointing this out. Following your comments, I made changes to the article in the first paragraph on page 4 of the paper, marked with a green background]

Comments 2: [ The authors' motivation in conducting their research is unclear. What knowledge gaps do they want to fill with their article?]

Response 2:[ The  purpose of writing this article is to improve the dynamic positioning capability of the intelligent and practical training ship, and to use terminal sliding mode to improve the accuracy and robustness of the control system. And it fills the gap. Most of the ships studied in the dynamic positioning system are ship models or for-profit ships, which have many shortcomings. The use of intelligent and practical training ships is of great significance to the research of the dynamic positioning system.]

Comments 3: [ Unclear structure of the paper. Sections 1 and 2 are devoted to the description of the ship, not to the issues of dynamic positioning.]

Response 3: [Thank you for pointing this out. Following your comments, I made changes to the article in the first paragraph on page 4 of the paper, marked with a green background, I added an introduction to ship dynamic positioning and the work done by predecessors at the front of Chapter 3.]

Comments 4: [There are no conclusions. Instead of conclusions, a description of the figures is presented.]

Response 4: [Thanks for your suggestion, I revised my conclusion on page 19 of the article, marked with a green background]

Comments 5: [ Many blunders in the text. It is obvious that the article was made in a hurry.]

Response 5:[ I'm very sorry for this. This is my first time writing a paper, and I didn't finish it very well. I improved the deficiencies in the paper based on the opinions of other reviewers and you. Of course, if there are still deficiencies, I hope you can correct them , for which I am very grateful]

Round 2

Reviewer 2 Report

Comments and Suggestions for Authors I am very pleased with the thorough revisions made by the author. I look forward to an illustrious future for my research and anticipate outstanding academic achievements ahead. Thank you.

Reviewer 3 Report

Comments and Suggestions for Authors

I have carefully reviewed the revised version of the manuscript, and after reviewing all the changes made by the authors, I have found that they have responded positively to all my comments, and it now deserves to be published in the journal.